# Understanding shallow landslides in Campos do Jordão Municipality – Brazil: disentangle the anthropic effects from natural causes in the disaster of 2000

Rodolfo M. Mendes[1], Márcio Roberto M. de Andrade[1], Javier Tomasella[1], Márcio Augusto E. de Moraes[1], Graziela B. Scofield[1]

[1]National Center for Monitoring and Early Warning of Natural Disasters, Parque Tecnológico/São José dos Campos, Estrada Doutor Altino Bondesan 500,12247-016, São Paulo, Brazil

*Correspondence to*: Rodolfo M. Mendes (rodolfo.mendes@cemaden.gov.br)

**Abstract.** Located in a mountain area of Southeast Brazil, the municipality of Campos do Jordão has been hit by several landslides in recent history. Among those events, the landslides of early 2000 were significant for the number of deaths (10), the population affected and the destruction of infrastructure that caused. The purpose of this study is to assess the relative contribution of natural and human factors in triggering the landslides of the 2000 event. To achieve this goal, a detailed geotechnical survey was conducted in three representative slopes of the area to obtain geotechnical parameters needed for slope stability analysis. Then, a set of numerical experiment with Geo-Slope software was designed including natural and anthropic factors separately. Results showed that natural factors, thus is, high intensity rainfall and geotechnical conditions, were not severe enough to trigger landslides in the study area and that human disturbance were entirely responsible for the landslides events of 2000. Since the anthropic effects used in the simulations are typical of Brazilian hazardous urban areas, we concluded that the implementation of public policies that constrain the occupation of landslide susceptible areas are urgently needed.

## 1 Introduction

Due to the combination of frequent heavy rain of high intensities on landscapes dominated by narrow valley and steep slopes, large areas of the southeast and south of Brazil are naturally susceptible to landslides. In addition, recent studies by CEPED (2012) indicates that more than 160 million inhabitants live in urban areas (about 90% of Brazilian population), and that increased urbanization in cities without urban planning and consequent occupation into hazardous areas have led to the increasing of this natural disaster~~In addition, population growth, increased urbanization, recent estimations (CEPED 2012) indicates that more than 160 million inhabitants live in urban areas (about 90% of Brazilian population), and expansion of urban construction into hazardous areas have led to an escalating impact of this natural disaster~~. Consequently, landslides are directly associated with loss of lives, property and infrastructure damage, and environmental destruction. During 2011, for

instance, mountainous regions of Rio de Janeiro State suffered several landslides that caused more than 1,500 deaths and severe damage to the urban and rural infrastructures (Coelho Netto et al., 2013).

In spite of floods (generally gradual flooding) being the most disrupted natural disasters in terms of economic damages ad population affected, landslides have been considered the most severe in terms of dead toll (Londe et al., 2014). Although landslide and flashfloods usually affect heavily urbanized downtown areas, it is also recognize that poor population living in the outskirts are more vulnerable to these types of disaster. Increased landslide hazard, for instance, has been related to the improper cut-and-fill construction of self-built housing on steep slopes, after the removal of vegetation. In addition, because of the lack of collection systems, sewage is disposed into the hillslope soils further increasing the risks of triggering landslides, not to mention the health risks associated with the lack of sanitation.

Among those areas affected by landslides in SE Brazil, the municipality of Campos do Jordão has been hit by several events since the seventies. The most recently severe landslides occurred in the early days of 2000, leaving 103 people injured, 10 fatalities, and more than 423 houses at risk of collapse (Londe et al., 2014).

Early warning system used by the civil defense in Campos do Jordão and by CEMADEN (National Center for Monitoring and Early Warning of Natural Disasters) are based on threshold values of 72 h accumulated rainfall, derived from empirical studies (Tatizana et al., 1987; Santoro et al., 2010). In 2000, rainfall was monitored every 24 h at 7:00 am using manual rain-gauges, and the threshold value for triggering landslides were based on previous studies in other areas of Brazil. Since the accumulated rainfall values responsible for the occurrence of the landslides of 2000 were well below the critical line proposed in previous studies (Tatizana et al., 1987), it was not possible to conduct the pre-emptive evacuation of many hazardous areas.

Although empirical rainfall thresholds are successfully used in operational warning systems to predict shallow landslides (Lagomarsino et al., 2013), critical rainfall thresholds for triggering landslides vary due to regional and local precipitation distribution, slope morphometry, soil characteristics, lithology, microclimate and geological history (Crosta, 1998; Van Asch et al., 1999). Therefore, the reliability of empirically derived critical rainfall threshold depends entirely on the availability of a significant number of cases relating the occurrence of landslides and rainfall conditions. In this sense, Guzzetti et al. (2007) lists as regional thresholds those covering regions with a few to many thousands square kilometers and having similar meteorological, climatic and physiographic characteristics, whereas for local conditions the geomorphology and climate regime are considered to be applicable to areas that range in the order of hundreds of square kilometers.

After the event of 2000, new critical 72 h rainfall thresholds values for landslides were proposed for the area. In addition, the Civil Defense of the State of São Paulo established new critical rainfall amounts for visually monitoring critical areas in order to detect early signals that may indicate the imminence of a landslide. However, the peak rainfall intensities recorded in the event of 2000 did not repeat even since, while irregular occupations continued in many landslide prone areas. Therefore, a detailed study of the event of 2000 is relevant not only because it's extreme characteristics, but also with the perspective that memory of the 2000 event is dim and its impacts is largely underestimated among many local residents.

In this context, for a limited number of data hydrologic models are relevant for investigating precipitation induced shallow
planar landslides (Terlien, 1998).
Given the lack of detailed data from historical landslides events in the municipality of Campos do Jordão, the aim of this
study was to understand the factors responsible for triggering the landslides of early 2000 in the area using a numerical
model that fully couple slope stability analysis with saturated/unsaturated transient pore-water pressure simulations.
Physical-based hydrological models have been widely applied to predict pore-pressure build-up due to the infiltration in
shallow landslides (Frattini et al., 2009; Iverson, 2000). Several models, based on the infinite slope concepts, that integrates
hillslope hydrology with slope stability, are reported in literature: for instance SINMAP (Pack et al, 1998), SHALSTAB
(Dietrich et al., 1998), TRIGRS (Baum et al., 2002) and GEOtop-FS (Rigon et al., 2006).
During the last decade, physically based landslide prediction models have also been successfully used in early warning
systems. Models used in such applications include, among others, the Combined Hydrology and Stability Model-CHASM
(Thiebes et al, 2014), the High Resolution Slope Stability Simulator – HIRESS ~~HIRELESS~~ (Rossi et al, 2013); the SLope-
Infiltration Distributed Equilibrium-SLIDE (Liao et al, 2010; Montrasio and Valentino, 2008), the Shallow Landslides
Instability Prediction-SLIP (Montrasio 2000; Montrasio et al, 2011). Another slope stability model is the modular software
package GeoStudio (2012), in which SEEP/W and SLOPE/W plugins are used to simulate the instability of slopes during
extreme rainfalls. Although GeoSlope is a simplified "single slope" model, it has been used in several previous studies to
understand the effect of infiltration on rainfall-induced landslides (for instance Ng and Shi, 1998; Gasmo et al., 2000; Kim et
al., 2004; Huat et al., 2006; Oh and Vanapalli, 2010; Acharya et al., 2016), producing very good results (Tofani et al., 2006).
In this study, we analyzed several scenarios that included the relative influence of natural and anthropic factors that prevails
in the area, and identified the most critical factors responsible for the severe landslides of 2000 using the GeoStudio (2012)
software due to its versatility of for handling natural and anthropic boundary conditions separately.
In addition, we analyzed if the threshold rainfall values establish by the Civil Defense are adequate for early warning of
landslides occurrence taking into account the today´s occupation patterns of landslide prone areas.

## 2 Material and Methods

### 2.1 Study site

The study site is the municipality of Campos do Jordão, of the State of São Paulo, located in a mountainous region along the
Mantiqueira Hills (Figure 1). In geological and geomorphological terms, the Campos do Jordão plateau is a crystalline
plateau block with elevations of more than 2,000 m above sea level and bordered by steep cliffs that rise approximately
1,500 m over the adjacent Paraiba valley (Almeida, 1976). The relief, strongly conditioned by the structures and lithologies
of the area, is characterized by the presence of high hills and erosion of grandstands. On the basis of these amphitheaters
occur peat depressions (Modenesi-Gauttieri and Hiruma, 2004), where deposits of organic clay of varying thickness are
found. The geological and geotechnical characteristics of the deposits of organic clay and its quite sensitive behavior to
sudden human interventions that alter their original equilibrium conditions have conditioned the slopes stability in the urban
area of the municipality of Campos do Jordão (Ogura et al., 2004).
The area where Campos do Jordão is located was occupied by Portuguese settlers during the XVIII century. During the
hygienist movement (late XIX and early XX centuries), various health facilities were established in the town, mainly for
tuberculosis treatment. Since 1940, the town experienced a large population growth and urban expansion due to the
development of tourism: the number of inhabitants increased from 13 thousand in 1950 to more than 50 thousand according
to the estimates, with density of 164.76 pop/Km$^2$, 99.3% of which live in the urban area (IBGE, 2016).
The process of accelerated urbanization, specially from the 70s, of areas with unfavorable geotechnical characteristics, has
been pointed out as responsible for most of the natural disasters in Campos do Jordão (Ridente et al., 2002). Table 1 shows
the most important events in terms of dead toll and damages recorded in the area.
Landslides in the study area are classified as shallow, translational type, with depths of the rupture surfaces less than 2 m.
Depending on the position of the rupture, three different processes are observed: the rupture surface occurs in the residual
soil of undisturbed ground; the rupture surface occurs in the residual soil of a slope cut; and the rupture surface occurs in the
base of the landfill deposit, or in the slope residual soil with mobilization of the overlying landfill. The last landslide types
are more harmful since they mobilized larger amounts of material. In the case of the event of 2000 (Figure 2), which is the
focus of this study, rainfall began on 31/12/1999, and continued almost uninterrupted for 4 days with high intensity rainfall
bursts. According to the Brazilian Center for Weather forecasting and Climate Study – CPTEC, daily rainfall from
31/12/1999 through 05/01/2000 was, respectively, 78.5, 101, 120, 60, 144.5 and 10.5 mm (Ridente et al., 2002). Landslides
associated with this event, were considered to be one of the most severe in urban areas in Brazil, since hundred of landslides
occurred, mostly in slopes in poor neighborhoods where houses are constructed over cut-and-fill areas.
Based on the landslides events of 1972, 1991 and 2000, Ridente et al. (2002) proposed an approximation of the critical
rainfall necessary for the deflagration of landslides in Campos do Jordão, revealing that, in most cases, landslides are due to
occur after three-day rainfalls of about 200 mm, with a daily rainfall of at least 70 mm during the last day analysed. The
Civil Defense Preventive Plan of the State of São Paulo uses three indexes of precipitation accumulated in three days (60, 80
and 100 mm) as critical thresholds to enter into warning level (Santoro et al., 2010). These thresholds were based on the
studies carried out by Tatizana et al. (1987) and has been considered a critical value for issuing early warnings based on
rainfall observation and forecasting.
Aiming to develop relationships for the prediction of mass movements in the area, Ahrendt (2005) attempted to correlate
precipitation with the occurrence of landslides based on the critical intensity curves obtained by Tatizana et al. (1987) for
Serra do Mar in the municipality of Cubatão, and by D'Orsi (1997) for Serra da Mantiqueira in the Municipality of Rio de
Janeiro. Results showed that the occurrences of Campos do Jordão were below the critical lines of those areas, indicating
that the rainfall intensities required for triggering landslides in Campos do Jordão are much lower if compared to the other
sites. Therefore, the study of Ahrendt (2005) concluded that rainfall characteristics that triggers landslides in Campos do
Jordão are very unique and a different and more detailed approach was needed.
Considering the limited historical data of landslide occurrences and the few previous studies in the area that makes extremely
difficult to define accurate critical threshold rainfall values that triggers landslides, in particularly, the effects the
accumulated rainfall on the water movement and its relationship to rapid mass movements.
The Brazilian National Centre for Monitoring and Early Warnings of Natural Disasters – CEMADEN began to monitor
Campos do Jordão by the summer of 2012. Most of the occurrences were observed in cut-and-fill slopes, with evident
contribution of wastewater and micro drainage deficiency. Recent history did not recorded occurrences of great magnitude;
however, the destruction, or even the prohibition of occupying damaged houses, is a recurrent problem.

**2.2 Soil moisture monitoring**
Soil moisture was monitored in every hourly interval to a depth 3.0 meters along the 12 months (01/01/2016 to 12/31/2016),
using two EnviroScan$^{TM}$ probes installed next of the borehole SD-03 (Figure 3). EnviroScan$^{TM}$ probes are installed into
customized access tubes manufactured by Sentek Pty. Ltd. Inside of the EnviroScan$^{TM}$ probe were distributed six Sentek
capacitance sensor. The capacitance sensor gives an output in volumetric water content (mm of water per 100 mm of soil
measured). This is converted from a scaled frequency reading using a default calibration equation, which is based on data
obtained from numerous scientific studies in a range of soil textures.
Before a Sentek capacitance sensor can be installed in the soil, it must have minimum and maximum values set. This is done
using air and water around each sensor (lecture limits of the volumetric water content – dry and saturated, respectively).
Soil moisture was monitored during 2016 at hourly intervals and to a depth of 3.0 m using two EnviroScanTM (Campbell
Scientific, 2016) probes installed next of the borehole SD-03 (Figure 2). Each probe included six capacitance sensors that
measured soil moisture every 0.5 m, thus is, at the depths of 0.5, 1.0, until 3.0 m deep, which allowed to monitor moisture
variations of the landfill, residual and saprolite layers. Before the EnviroScanTM capacitance probes were installed in the
soil, maximum and minimum values were normalized by matching the raw readings from each sensor at both 0% (held in
air) and 100% water levels (submerged in water).
**2.3 Geotechnical survey**
SPT (Standard Penetration Test) boreholes were drilled along three profiles of the study site (A-A'; B-B'; C-C' in Figure 3) at
six different positions along the slopes (SD-01 to SD-6, Figure 4). Disturbed and undisturbed samples were taken from the
boreholes for the determination of the parameters used for stability analysis.
Three (03) undisturbed samples were collected in migmatitic saprolite block close to the SD-04 borehole. This material
occurs anisotropically and discontinuously, because it presents significant textural variation resulting from the heterogeneity
of the parental rock, being predominantly formed by silt and fine sand, with variable occurrence of clay. From the 6
boreholes (SD-1 to SD-6) and the 3 undisturbed blocks it was possible to obtain a total of 12 soil samples to perform
geotechnical characterization tests of the study area following Brazilian standard procedure.
Disturbed and undisturbed samples collected were used to perform grain size analysis test (ABNT, 1984b and 1995), soil
particle density (ABNT, 1984a), bulk density, specific dry mass and Atterberg limits (ABNT, 1984c and 1984d). Parameters
of effective cohesion (c') and effective friction angle ($\phi$') were obtained from saturated direct shear tests, using square
shaped undisturbed samples with 60 mm of side and height of 25 mm. The soil samples were in their natural state, being
representative of the "Residual" and "Saprolite" soil layers. During the consolidation step, all specimens were saturated for
24h and subjected to net normal stresses of 25, 50 and 100 kPa. Then, in the shearing phase, a constant velocity of 0.033 mm
min$^{-1}$ was applied. Vertical and horizontal displacements were recorded during the consolidation and shearing phases.
After saturation soil samples for 12 hours, Water Retention Curves - WRC of the residual soils layers were obtained using
pressure plate for suctions <100 kPa and filter paper for suctions $\geq$ 100 kPa for the drying path of the samples following the
recommendation of Marinho and Oliveira (2006). Results showed that the differences of water retention values at the
transition among both method were not significant, making unnecessary further adjustments (Figure 8). The saturated
hydraulic conductivity - Ksat was obtained in laboratory using a constant head permeameter. Hydraulic conductivity
functions were estimated from the WRC, Ksat using the Van Genuchten (1980) model. In the case of landfill deposits, the
values of Ksat for different soil texture were those obtained Ahrendt (2005) from core measurements.

**2.4 Modelling experiments**

The modelling of the stability and seepage analysis was divided in two parts: (1) transient unsaturated seepage analysis; (2)
stability analyses coupled with the results from the previous step.
For the seepage analysis, 35-days accumulated rainfall of the period 01/12/1999 through 04/01/2000 was considered, since
that event triggered several landslides in the study area (Ahrendt and Zuquette, 2003). In addition to the geotechnical
parameters, anthropic factors that induce landslides typical of Brazilian urban slopes, specifically housing load, man-made
cuts and leakage from pipes, were included in the modelling experiment with the aim of analysing the degree of influence of
these factors on the trigger of landslides in the study area during 2000 (Figures 4, 5 and 6).
The boundary conditions were set according to field observations on the landslide area and boundary conditions used by
Rahardjo et al. (2007). The non-saturated transient flux results were obtained for two cases: considering only the
accumulated rainfall and rainfall including linear leakage along the cut slope (Figure 5). The initial pore-pressure values used
in the transient flow analysis were obtained indirectly from the WRC and the data of the soil moisture sensors installed in the
study area (Figures 8 and 7). Next, the factor of safety (FS) for the slope was estimated from the transient seepage modelling
coupled with the stability analysis tool (Geo-Slope, 2012a). All the stability analyses were conducted considering the theory
of static equilibrium of forces and momentum. The FS were calculated using the geotechnical and anthropic parameters,
obtained from the method of Morgenstern-Price, which considers circular and non-circular rupture surfaces. All the
simulations allowed the slope stability module SLOPE/W to identify the most critical rupture surface (Figure 6). Therefore,
the values of the Slope Safety Factor – FS, were the lowest of all conditions analysed.

**3 Results**
**3.1 Geotechnical survey**
The result of the granulometric analyses of the residual and saprolite layers of the three profiles studied are presented in
Figure 7. Residual layer (sample SD-01/2.0 m) can be classified as clayey sand, with percentage of sand and silt of 53 % and
25 %, respectively. The soil samples representative of the saprolite layer showed a significant variation of the percentage of
the clay fraction (3 to 24%), silt (14 to 42%) and sand (53 to 73%), indicating that soil profiles are heterogeneous, which in
agreement with the textural characteristics of its parent material (migmatitic gneiss). Therefore, it is expected that the
mechanical and hydraulic properties of this soil layer present high variability. The general results of the geotechnical tests
(general characteristics, shear strength and saturated hydraulic conductivity) of the samples of the representative soils of the
studied area are presented in Table 2.
Analysing the values of the effective strength parameters (c' and $\phi$') and saturated hydraulic conductivity (Table 2), the
values representative of the 'Saprolite' layer showed significant variability: the coefficient of variation was 85% for the
effective cohesion; 20% for the effective friction angle, and 89% for the saturated hydraulic conductivity, reflecting the
heterogeneity character of the parent material. The high values of the resistance parameters shown in Table 2 can be
explained are associated with the high heterogeneity of the residual gneiss soil, such as the presence quartz particles and
other minerals of considerable size in the specimens tested, which confer them high resistance. In addition, the values of the
resistance and Ksat parameters obtained in this study are close to mean reference values of residual gneiss soils
representative of other Brazilian sites (Costa Filho and Campos, 1991; Ahrendt, 2005; Reis et al., 2011).
Figure 8 shows the water retention curves of the soil layers representative of the profiles. In general, the residual and
saprolite layers are able to hold more water compared to the landfill deposit. For example, for a field matrix suction level of
100 kPa, the volumetric moisture values of the Landfill Deposit, Residual Soil and Saprolite layers are, respectively, 0,06
$m^3/m^3$; 0,24 $m^3/m^3$; 0,26 $m^3/m^3$.

**3.2 Soil moisture data**
Soil moisture data from 2016 from the EnviroScanTM probes (3G1 and 3G2) are presented in Figure 9. The data from 3G1
(upper graph of Figure 9) showed that the sensor installed at a lower depth (0.5 to 1.0 meter), representative of the landfill
deposit (green curve) layer, have variations larger soil moisture variations ($\Delta\theta = 32\%$), with maximum and minimum water
content values recorded in March (46%) and April (14%), respectively. At deeper layers (1-3 meters deep), representative of
the "Residual" and "Saprolite" layers (black and red curves), time variation of soil moisture is much lower ($\Delta\theta = 10\%$, on
average). In the Residual layer (black curve), maximum and minimum values of soil moisture were verified in January
(38%) and May (27%), respectively. In the case of the saprolite layer (red curve) maximum and minimum humidity values in
the months of June (40%) and May (32%). The different dynamics among the three soil layers are reflecting not only
differences in the retention properties of each layer considered but also the deep soil water dynamics down the soil profiles.
This explains why the upper layer (landfill deposit) shows a more spiky behaviour in response the rainfall; while the other
two layers exhibit gradual and delayed variations related to deep water percolation.
Analysing the data of the probe 3G2 (lower graph of Figure 9), it is clear that the soil moisture variation of the sensor of the
surface layer, representative of the Landfill Deposit layer (green curve), was significantly lower ($\Delta\theta$ = 15%) compared to the
same layer of probe 3G1. Maximum and minimum soil moisture values were recorded in September (43%) and August
(28%), respectively. At deeper depth, within the residual and saprolite layers (black and red curves), time variation of soil
moisture variation is similar than the measurements of probe 3G1 ($\Delta\theta$ = 9%, on average). For the "Residual" layer (black
curve) maximum and minimum soil moisture values were observed in January (31%) and May (21 %), respectively; while in
the "Saprolite" layer (red curve) maximum and minimum soil moisture values in the months of January (37%) and August
237 (30%).
Contrasting differences in the soil moisture behaviour of the landfill deposit from the probes 3G1 and 3G2 suggest that the
variability of soil parameters is higher in the top layer. This was expected considering that this layer is the result of the cut-
and-fill processes mixed with construction wastes of several types.

**3.3 GeoSlope simulations input data**

Table 3 summarizes the parameters used in the numerical simulation with GeoSlope software, based on the geotechnical
survey and information extracted from different sources.
Regarding the geotechnical properties, in order to reduce the uncertainties due to the heterogeneity of the parent material, the
mean values of the resistance (c' and $\phi$'), bulk density and saturated hydraulic conductivity parameters for saprolite and
residual (Table 2) were used in the flow and stability modelling. As mentioned before, for the landfill deposit, the
geotechnical parameters were those obtained by Ahrendt (2005).
Based on the field information from previous studies in Brazil, the anthropic factors considered in the simulations were:
point leakage sources of 1.0 m$^3$day$^{-1}$ (SABESP, 1993 and 2016) for simulation that include leakage; distributed load due to
one floor housing of 2.0 kNm$^{-2}$ (ABNT, 1980); height of the cutting slope (based on field data). In the case of the simulations
that include leakage, one point of leaking constant value of 1.0 m$^3$day$^{-1}$ was considered in the cut slope, after the 10$^{th}$ day of
the simulation until the 35$^{st}$ day, since numerical experiments showed that a time interval of 10 days was adequate to
minimize the effect of the uncertainties of initial pore-pressure conditions used in the simulations. It should be noted that this
strategy helped to separate the effect of leakage from other anthropic factors, without impacting the results at the end of the
simulation period (day 35).
For the transient flow analysis, the highest average moisture values of the three layers was considered (Landfill deposit -
$\theta_{mean}$ = 33%, Residual - $\theta_{mean}$ = 31%, Saprolite - $\theta_{mean}$ = 34%, to November 30, 2016 - light red region in Figure 9).
Subsequently, the mean values of humidity were used to obtain the initial values of matrix suction from the representative
water retention curves of each layer of the profile. The proper choice of the initial values of matrix suction is to provide the
numerical model with a fast and coherent convergence in the pore-pressure distribution of water in the soil layers, aiming to
adjust them satisfactorily to the rain data considered in the flow analysis (01 From December 1999 to January 4, 2000 - 35
days).
For the transient flow analysis, the initial conditions of the simulations were derived assuming initial soil moisture values of
0.33 $m^3.m^{-3}$, 0.31 $m^3.m^{-3}$ and 0.34 $m^3.m^{-3}$, for the landfill deposit, residual and saprolite layers respectively. These values
correspond to average of the highest soil moisture values of the two probes during November 2016, and indicated by the
light-red shaded area of Figure 9.
Subsequently, the mean moisture values were used to obtain the initial values of matrix suction from the representative water
retention curves of each layer of the profile (Figure 8). This choice of the initial values of matrix suction of the numerical
experiments proved to be crucial to achieve a fast and coherent convergence of pore-pressure distribution of soil layers in the
simulations, since they are representative of the 35 days period considered in the flow analysis (From 01/12/1999 through
04/01/2000). Based on this approach, the values of negative pore pressure (matrix suction) were -10 kPa for landfills, -40
kPa for residual and -40 kPa for the saprolite.

**3.4 Slope Safety Factor Analysis**
Figure 10 shows the time variation of the slope safety factor (FS) during 35 days for the 2000 rainfall for the three profiles.
In Figure 10, two "warning zones" are considered: zone of instability FS <1.0, where ruptures should occur; and low stability
~~stable~~ zone, 1.0 <FS <1.5, which indicates a low possibility of landslide occurrence. In this warning zone the Brazilian
Association of Technical Standards (ABNT, 1991) established the following conditions for the safety degree of the slope:
High (1.3≤FS <1.5); Mean (1.15≤FS <1.3); Low (1.0≤FS <1.15).
For all three profiles analysed, it is clear that the effect of daily rain on the decrease of the FS (green line) was practically
insignificant, with FS values above the 1.5 threshold (high safety degree of the slope); indicating very low likelihood of
landslides. In the "rainfall only" scenario, the variations in FS values are due to the geotechnical and geomorphological
characteristics of the analysed profiles only. For the "rainfall only" scenario, the difference between FS values in the three
profiles are mainly due to: the surface slope, since the profile AA´ is steeper than CC' which is steeper than BB'; due to
differences in the thickness and location of the layers along the slope (Figure 4) and; the water table profile, related to the
soil layers.
For the second scenario considered in the analysis of the stability, which includes cut-and-fill effects besides rainfall (red
line in Figure 10), it was verified that the effect of terrain cuts caused minor effects in the slope safety factor. Except for the
case of the A-A' profile, which presents the FS condition <1.5 between 32th and 35th day after the beginning of rainfall, FS
values were above the 1.5 threshold. However, it is important to note that, in the case of the profile A-A', the decrease of FS
was more pronounced than in the other profiles analysed, directly related to the positioning of the cuts considered along the
slopes located based on field information. The configuration of the cuts used in this profile favour the wetting of the top soil
layer and, consequently affected the whole profile stability.
The third scenario of Figure 10 (black line), where the joint influence of two anthropic factors (cut and leakage) with the
rainfall of 2000 was considered, and the variations in FS values are significant. For the profile A-A', FS values remained
below the threshold of 1.5; while in the other two profiles FS dropped below 1.5 between the 32nd and 35th days of
simulations in the case of the B-B' profile, and on day 17th for the C-C' profile. It should be noted that, after the 11th after
the beginning of simulation, FS values become sensitive to rainfall variability.
In addition, it can be seen in Figure 10 that the profiles B-B' and C-C' showed greater sensitivity to leakages, mainly due to
the geological-geotechnical characteristics, and the location of the cuts along the slopes, that favoured the decrease of the
matrix suction values and, therefore, induced instability in both profiles. In addition, critical condition, FS <1.0, in profiles
A-A' and C-C' are verified between days 32nd and 35th after the beginning of simulations, in response to the significant
rainfall that occurred at the period. However, it is observed that, under the influence of leakage, previous rainfall history
played a role since the factor of stability is lower previously to the large rainfall event of the end of the simulation period.
This can be seen in more clearly in the profile B-B' of Figure 10: in the dry period between day 15 and day 22 after the
beginning of simulation, it is verified a quick recovery of stability in the simulations that includes the effect of leakage
(black curve), which is interrupted with the return of the rainfall.
Finally, in the fourth scenario of Figure 10 (light brown line), when all the anthropic factors (cut, leakage and housing loads)
are considered together with the daily rainfall. For most of the time, FS values remained below the threshold of 1.5 in in the
A-A' and B-B' profiles, while in the case of the profile C-C´ only after the day 34th of the simulation. The probability of
landslides increased significantly (FS <1.0) in all profiles from day 32 of the simulation in response to heavy rainfall at the
end of the period. In the case of the profile C-C' (light brown line), the inclusion of housing loads appears to provide more
stability to the profile, probably related to the fact that the critical failure surface estimated by the numerical model was
different from that assumed in the case scenario 3 (black line).
Based on the assessment of the slope safety factor presented in Figure 10, it is clear that the probability of landslides
associated to 2000 rainfall on slopes covered with natural vegetation is very low. When considering the influence of rainfall
in conjunction with anthropogenic factors, there was a significant decrease in the safety factors in all the profile studied,
although the effect varied between slopes depending on the geological-geotechnical profile characteristics, geomorphological
conditions, water table position and the anthropic conditions, thus is, the positioning of cuts, leakage and housing loads along
the slope.
In general, slopes became unstable (FS<1.0) between 32nd and 35th after the beginning of simulations when high daily
accumulated values were verified. Since most of the landslide occurred on day 32nd, it follows that the model successfully
predicted the time were the landslides began. However, it should be noted that previous accumulated rainfall values were
crucial to create favourable conditions for triggering landslides as shown by Figure 10 after the 30th day from the beginning
of simulation.
Santoro et al. (2010) recommended in-situ technical surveys of urban hazardous areas after accumulated 72-hour rainfall
equal to 60, 80 and 100 mm (depending on the municipality), in order to identify evidences of the imminence of landslides
and to enforce eventual preventive removal of population. It can be seen in Figure 10 that the 72h accumulated rainfall were
35 mm for the day 30 after the beginning of simulation; 35 mm for the day 31; 60 mm for the day 32; 191 mm for the day
33. Thus, critical rainfall thresholds (60-100 mm) were exceeded between 32nd and 33th day for the events recorded in the
year 2000.
In this period, the FS in the three analyzed profiles presented the lowest values, located exactly between the "low-medium
security zone" (1.0 <FS <1.3) and "unstable zone" (FS <1.0), which shows that the "anthropic and natural factors integrated
analysis method" proposed in this paper successfully predicted the beginning of landslides. Although the 72h rainfall
threshold value proposed by Santoro et al (2010) proved to be valid for the 2000 event, results of the simulation indicated
that the rainfall 30 days previous to the landslides was crucial to bring FS values closer to critical levels, indicating that
critical value presents limitations on slopes initially drier.
**4 Conclusions**
The Geo-slope model proved to be an efficient and useful tool to predict the landslide of Campos do Jordão municipality due
to the rainfall event of 2000 and allowed to disentangle the effects of cut-and-fill, construction practices and pipe leakage in
three representative slopes of the area. The use of numerical models that perform flow and stability analyses considering the
simultaneous influence of natural and anthropic variables showed to be accurate for the prediction of occurrences of
landslides on urban slopes.
Regarding the rainfall critical values use in early warning system by CEMADEN and the Civil Defense for the Campos do
Jordão Municipality, although adequate for the event of 2000, our study show that the previous rainfall history, in
combination with leakages, played a fundamental role to create favorable conditions for the occurrence of landslides. This is
related to the fact that leakages contribute to keep the soil profile closer to saturation at the beginning of the period of more
intense rainfall, and consequently the developing of positive pore-pressure conditions. In other words, the threshold currently
used for issue early warning would result in late alarms under initial drier soil conditions, at least in heavily disturbed
landscapes.
The results of the stability analyses confirmed the hypothesis that the occurrence of landslides in the study area cannot be
attributed solely and exclusively to the rainfall events of the year 2000, despite the significant accumulated values.
Therefore, numerical modelling results corroborated the fact that the occurrence of landslides was the combination of natural
and anthropic factors, with the decisive influence of the latter, thus is, due to the presence of several cuts along the slope
combined with load of constructions and leakage. Clearly, human interventions on natural slopes play a fundamental role in

triggering landslides in heavily populated steep slopes surrounding urban areas. Once shallow landslides in the study area usually occur in cut and fill slopes, the rupture surface size and the amount of material mobilized do not vary significantly among events. Therefore, the most useful information for an early warning system perspective is to know whether the value of FS is below 1.0, regardless how much below that threshold the slope safety factor is. Another relevant information is the timing of the landslide events, since such information is crucial to determine the rainfall thresholds for issuing an early warning. Therefore, information about the rupture surface size, which is essential for assessing potential damages, is beyond the scope of this study.

Considering that the pattern of land use and construction used in the simulations is representative of most of the neighborhood of Brazilian urban areas, the methodology used in this paper needs to be repeated and verified in other areas in order to establish more accurate critical threshold that trigger landslides. Moreover, since the prone to landslide areas of Campos do Jordão Municipality are not the most populated of Brazil compared, for instance, to the outskirts of several metropolitan areas, it becomes crucial to verify whether a mosaic of site-specific rainfall thresholds is needed in heavily occupied areas, rather a single regional threshold, as suggested by Segoni et al. (2014). In this context, this study demonstrated the using the slope safety factor is viable for determine more accurate rainfall threshold that trigger landslides, with direct impacts on the credibility of early warning systems, which relies in minimizing false alarms or premature/late warnings.

Although the results of this study have uncertainties mainly associated with the geotechnical parameters used in the flow analysis and slope stability, it is the first comprehensive analysis of the factors responsible for triggering landslide in Brazil that integrates field evidence, anthropic effects, geotechnical data and numerical simulation. Since simulations results indicated that the slope safety factor FS was sensitive to both geotechnical and anthropic factors, future studies of slope stability probabilistic analysis, which takes into account the wider range of parameters values that occur in the study area, are needed.

Finally, considering that this work have demonstrated that the anthropic factors are the main instability factors in urban slopes, it is essential that urban managers and planners promote public policies and enforce laws that restrict the occupation of landslide susceptible areas. Detailed surveys to identify prone to landslide areas are essential, since many urban areas of Brazil lack zoning of hazardous areas, which is essential to implement regulations. Besides this, educational campaigns regarding the adoption of better construction practices and reducing piping leakage will be helpful in already consolidated occupied areas.

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

Table 1 – Historical disasters in Campos do Jordão Municipality. Source: Ridente et al. (2002) and Andrade (2014).

| Process | Location | Year | Damages | Causes |
|---|---|---|---|---|
| Earthflow | Vila Albertina | 1972 | 17 fatalities<br>60 houses buried | Saturated soil (8m thick), loading and vibration due to construction activities |
| Landslide | Britador, Vila Santo Antonio and Vila Paulista Popular | 1991 | 149 affected<br>11 houses buried<br>4 injured | 214.5 mm of rainfall in three days |
| Landslide and Mudflow | Britador, Vila Albertina, Vila Santo Antônio, Vila Nadir Vila Sodipe and Vila Paulista Popular | 2000 | 10 fatalities<br>1840 affected | 453.2 mm in five days |



Table 2 – Results of the geotechnical survey of soils of the study areas.

| Sample | Depth (m) | Soil layer | USCS | Unit weight (kN/m$^3$) | Effective cohesion (kPa) | Effective friction angle (°) | Hydraulic conduct. (m/s) | Gravel (%) | Sand (%) | Silt (%) | Clay (%) | $w_L$ (%) | $w_P$ (%) | IP (%) |
|---|---|---|---|---|---|---|---|---|---|---|---|---|---|---|
| SD-01 | 2.0 | R | SC | 18.3 | 37 | 56 | $4.44\,e^{-6}$ | 10 | 53 | 25 | 12 | 27 | 18 | 9 |
|  | 4.6 | S | SC | 19.1 | 18 | 37 | $9.46\,e^{-6}$ | 0 | 53 | 35 | 12 | 35 | 22 | 13 |
|  | 6.6 | S | SC | 17.9 | 2 | 49 | $7.93\,e^{-6}$ | 0 | 59 | 27 | 14 | 29 | 20 | 9 |
| SD-02 | 2.6 | S | SC | 21.4 | 19 | 34 | $1.18\,e^{-6}$ | 5 | 50 | 21 | 24 | 28 | 17 | 11 |
|  | 4.6 | S | SM | 17.5 | 14 | 42 | $3.76\,e^{-6}$ | 0 | 73 | 14 | 13 | 33 | 20 | 13 |
| SD-03 | 1.6 | S | SM-SC | 18.1 | 22 | 43 | $5.25\,e^{-6}$ | 1 | 59 | 29 | 11 | 22 | 15 | 7 |
|  | 2.6 | S | SM-SC | 16.8 | 2 | 52 | $6.13\,e^{-7}$ | 5 | 55 | 30 | 10 | 23 | 17 | 6 |
| SD-05 | 12.8 | S | SC | 17.5 | 48 | 54 | $2.77\,e^{-7}$ | 0 | 55 | 33 | 12 | 32 | 21 | 11 |
| SD-06 | 7.6 | S | SM | 17.8 | 42 | 28 | $3.09\,e^{-6}$ | 1 | 72 | 15 | 12 | - | - | - |
| Block-1 | 2.0 | S | SM | 16.0 | 2 | 53 | $9.37\,e^{-7}$ | 0 | 72 | 21 | 7 | - | - | - |
| Block-2 | 2.0 | S | SM | 16.0 | 49 | 37 | $2.98\,e^{-6}$ | 0 | 55 | 42 | 3 | - | - | - |
| Block-3 | 3.0 | S | SM | 16.0 | 13 | 46 | $4.44\,e^{-6}$ | 0 | 58 | 28 | 14 | - | - | - |

# Residual soil (R); Saprolite (S).





Table 3 – Geotechnical and anthropic parameters used on unsaturated seepage and stability analysis

| | | | | Geotechnical | | | | | | Anthropic | | |
|---|---|---|---|---|---|---|---|---|---|---|---|---|
| Profile | Slope declivity (°) | Slope height (m) | Soil layers | Shear strength effective parameters# | Unit weight (kN/m$^{-3}$)# | Rainfall (m/day) | Ksat (m/s) | Pore water pressure (kPa) | Level of the water table | Load in the slope (kN/m$^2$) | Height of the cut slope (m) | Leaking in the slope (m3/day) |
| A - A' | 11 - 40 | 130 | | | | | | | | | | |
| | | | Fill Deposit | c' = 2 kPa $\phi$' = 31° | 14,9 | Rainfall events of the 2000 year | 9.50 e$^{-6}$ | -7 | Data of SPT | 2.0 | 6.0 | 1.0 |
| B - B' | 16 - 33 | 95 | Residual | c' = 15 kPa $\phi$' = 36° | 18,3 | | 4.44 e$^{-6}$ | -40 | | | | |
| | | | Saprolite | c' = 21 kPa $\phi$' = 43° | 19,1 | | 5.43 e$^{-6}$ | -40 | | | | |
| C - C' | 15 - 35 | 87 | | | | | | | | | | |

# Fill Deposit (geotechnical parameters from Ahrendt, 2005)



















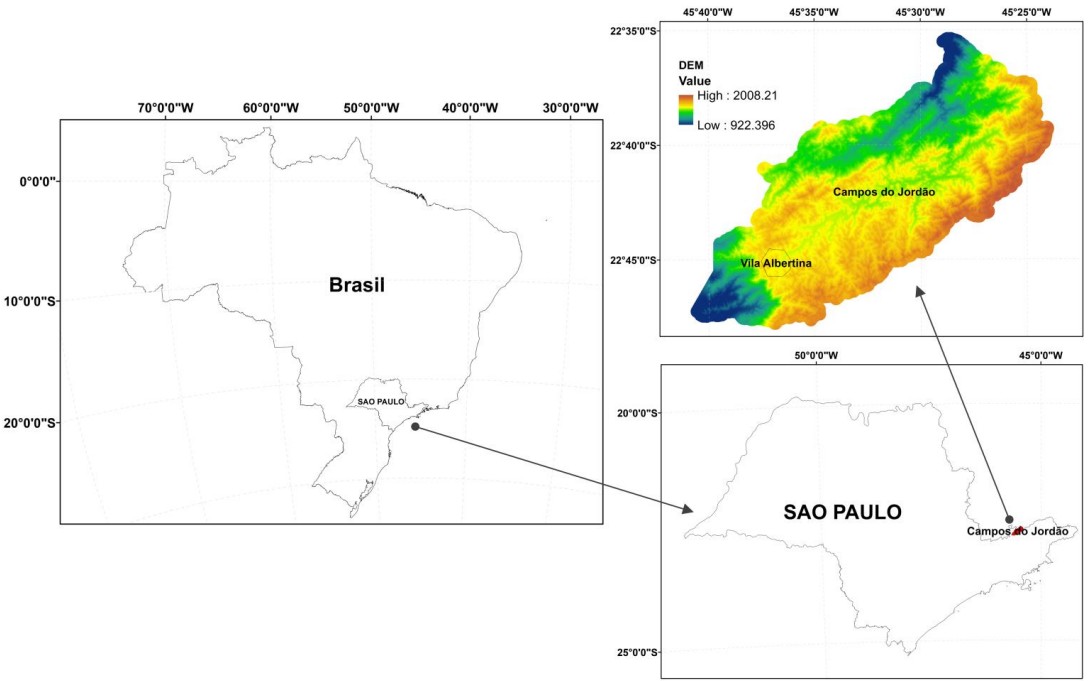

**Figure 1.** Geographical location of Campos do Jordão municipality and an inset of the study site.


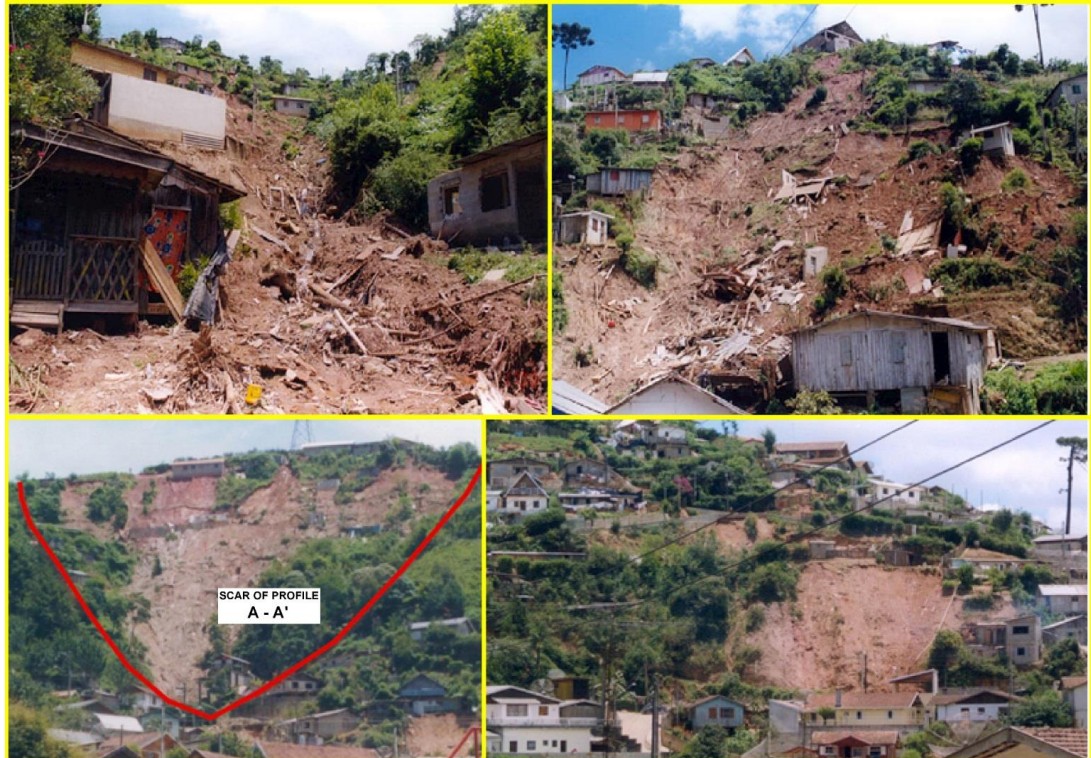

**Figure 2.** Shallow landslides that happened in 2000 on the study site (Source: Ridente et al., 2002)**.**


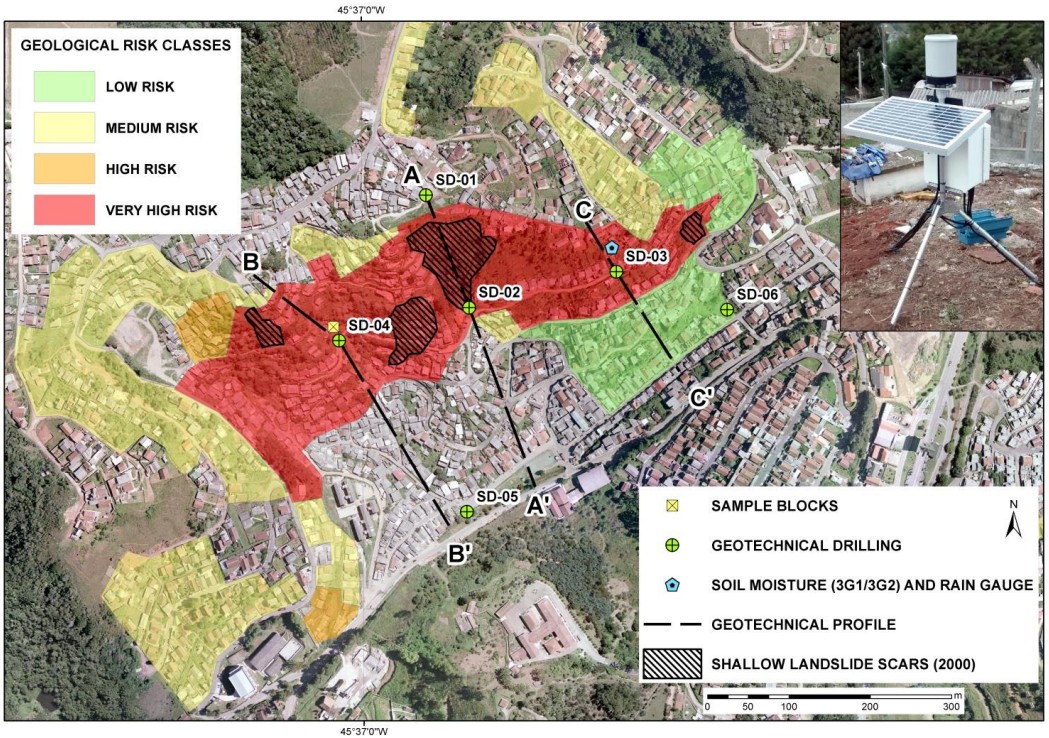

**Figure 3.** Satellite image of the study site showing the location of monitoring instruments (symbols), geotechnical transects (dotted lines
along the slopes); landslide susceptibility areas indicating the level of risk (areas shaded in yellow, orange and red); scars of previous
shallow landslides (black cross-hatched area).











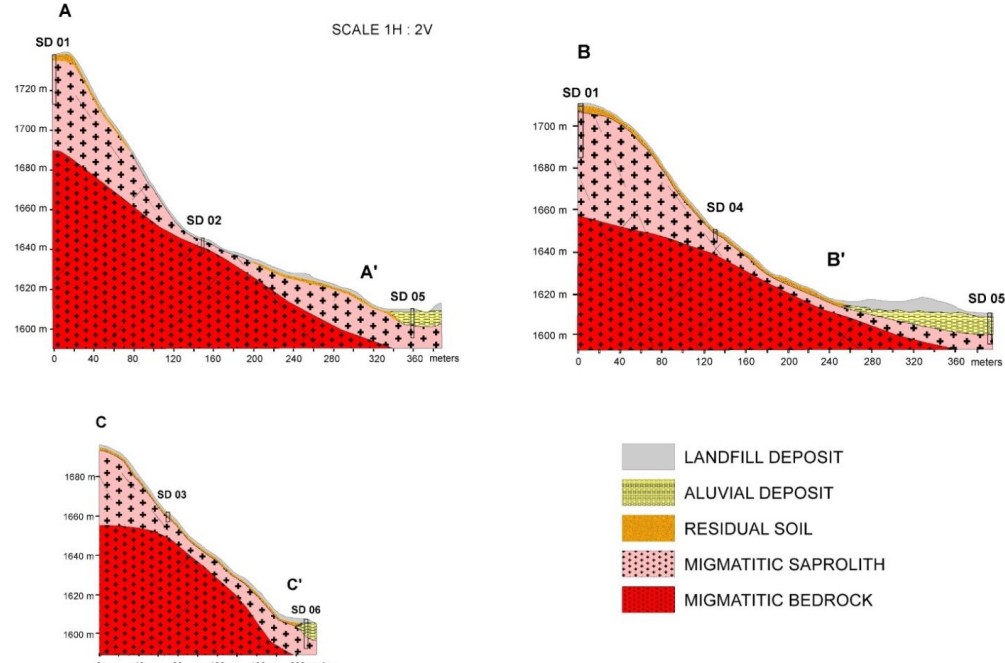

**Figure 4.** Geological-geotechnical profiles of the study area derived from the geotechnical survey.



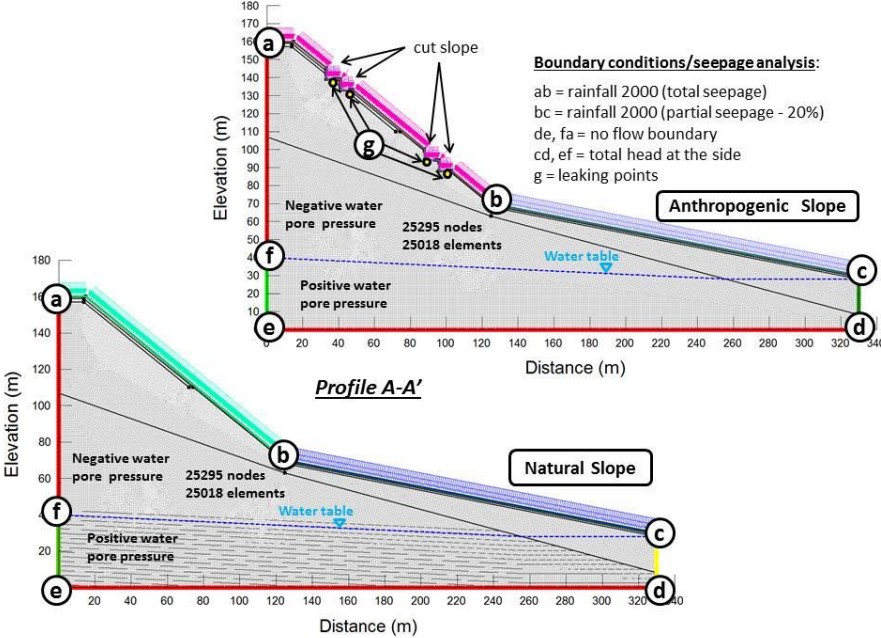


**Figure 5.** Slope geometry and boundary conditions used in the unsaturated transient seepage analysis considering natural and
anthropogenic factors (rainfall, cut slope and leakage).














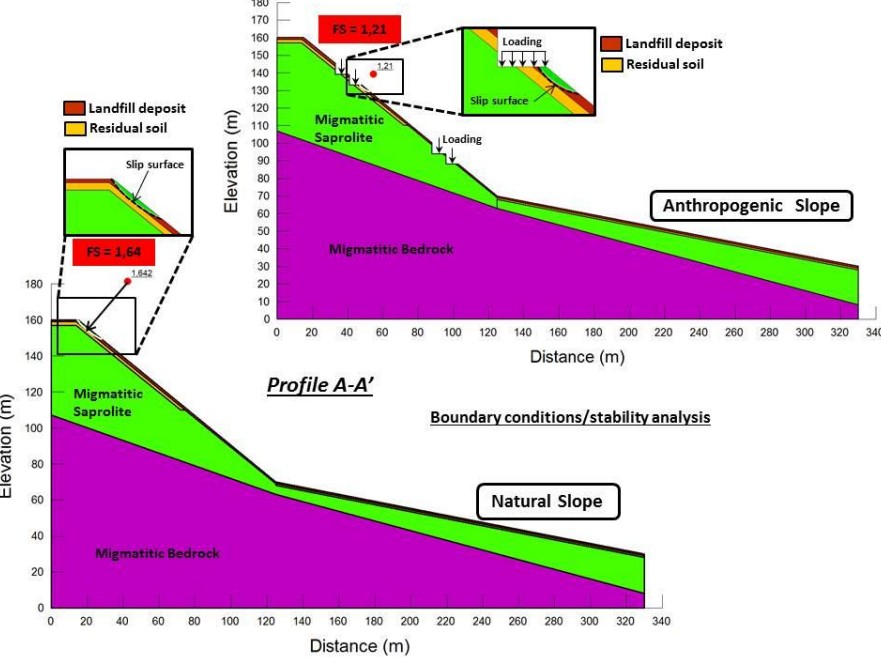


**Figure 6.** Slope geometry and boundary conditions used in the stability analysis considering natural and anthropogenic factors (rainfall,
cut slope, loading and leakage).














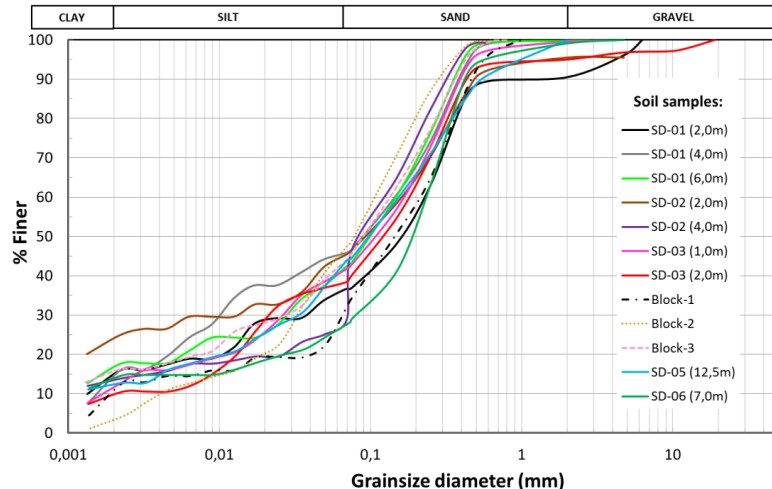


**Figure 7.** Granulometric distribution for the residual soil and the saprolite of the six boreholes analysed (SD-01 to SD-06) and for the undisturbed soil cores (Block-1 to Block-3).




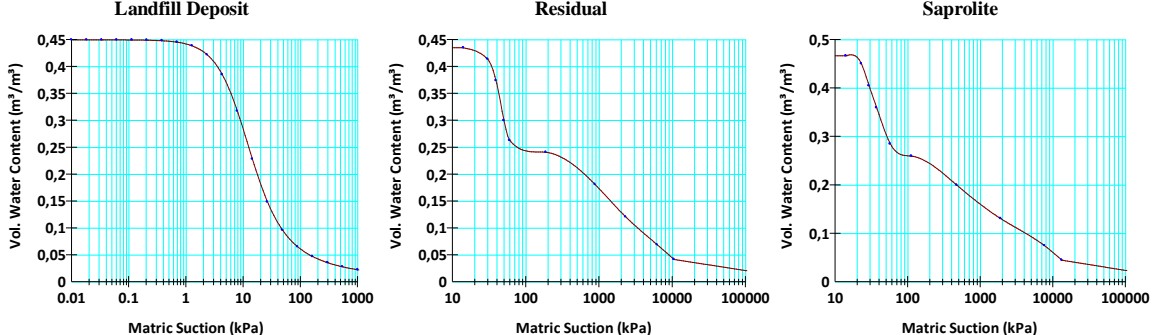

**Figure 8.** Water retention curves of the three soil types used in transient seepage analysis.




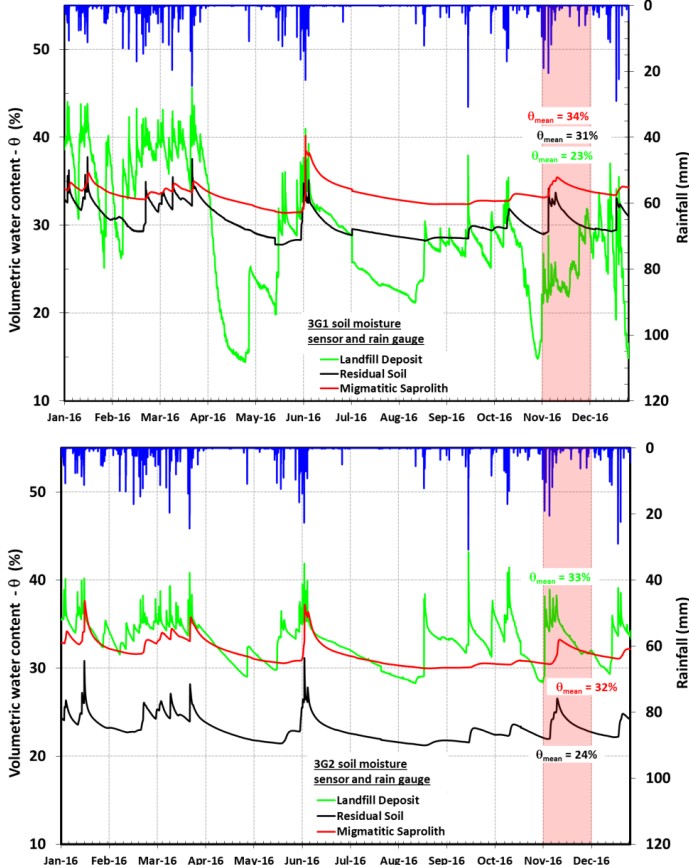

**Figure 9.** Time variation of soil moisture at different depths during 2016 in the study area in the sensor 3G1 (upper graph) and 3G2
664                                                      (bottom graph).








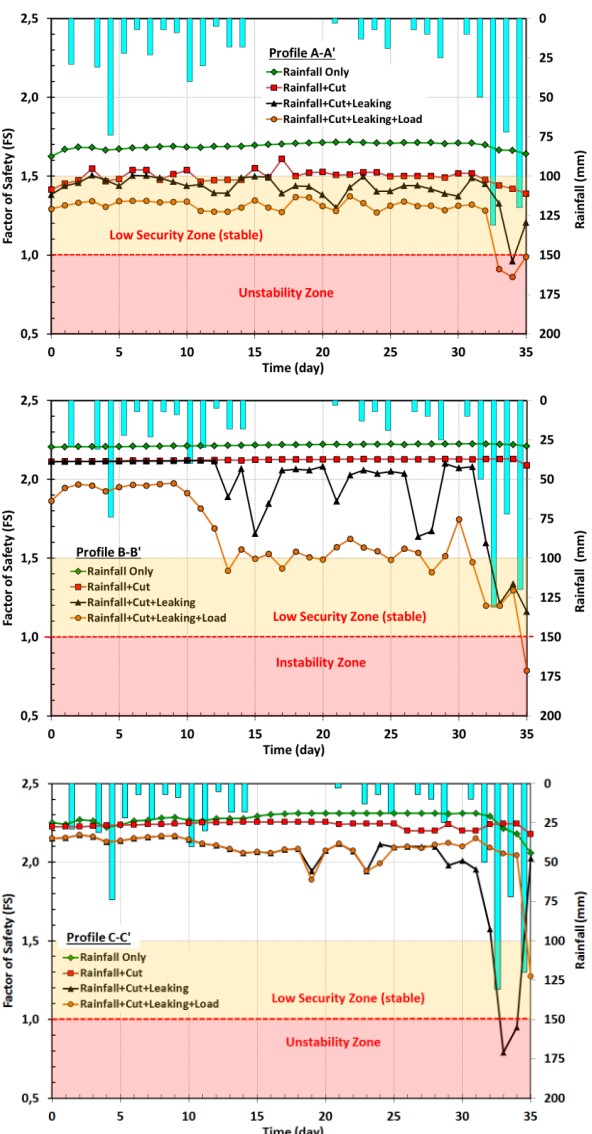

**Figure 10.** Time variation of the slope safety factor for natural conditions and taken into account the additional effects introduced by
anthropic disturbances on profiles A-A', B-B' and C-C'.