# Peer review of "Understanding shallow landslides in Campos do Jordão Municipality Brazil: disentangle the anthropic effects from natural causes in the disaster of 2000"

_Natural Hazards and Earth System Sciences, 2017_

## Referee Comment (RC1) · Anonymous Referee #1 · 17 Aug 2017

There is no doubt that the study has great importance associated to the monitoring systems of urban risk areas. In particular related to the use of precipitation data only. The combination of experimental laboratory data, field monitoring, geological aspects and the use and occupation of risk areas, brings to the study a set of information that deserves to be divulged. However, some aspects of the study and the text should be evaluated. The following are some comments aimed at improving and clarifying some points of the work. Line 30 - A reference on the given information of the percentage of the population that lives in urban area in Brazil would be important. Line 39 - The

reference to the removal of forests would not be removal of vegetation? Considering that it is already a deforested area. Line 66 – Typo: "Given the lack of detailed data" FROM…. Line 68 - The reference to numerical analysis is very broad. The numerical study was done with the flow analysis, but the stability analysis was done by equilibrium limit, which is not a numerical method. Line 93 – Figure 2 needs to be described in the text. Line 130 – Typo: Instead of lecture you mean reading? Line 132 to line 137 – This paragraph is repeated. Line 141 – Instead of Figure 3 it is Figure 4. The figure needs to be described in the text. Line 141 - The phrase seems truncated. The samples were not taken for stability analysis but for the determination of parameters that are used in the analyzes. Line 143 - The correct spelling is saprolite (saprolith seems to be Greek). This occurs along the text. Line 153 - The sentence is confusing and the order of the essay processes must be rewritten. For example: it is not the application of load that is made with a speed of 0.033mm / min, but the phase of shearing. Line 180 - The references to figures 3 and 4 are not clear. Where are the boundary conditions? Line 195 – It seems to be Figure 5 and not Figure 4. Line 195 – Why to use a "sandy loam" if the classification used is not from agronomy? Line 209 – Typo: "….soils representative of other Brazil….."?? Line 211 – Instead of Figure 5 it should be Figure 6. Line 213 - Why not use field capacity? Line 224 – The term humidity is used as water content. This does not seem correct. Line 224 – Will the article be published in color? Otherwise references to colors should be retracted from the text and another reference system should be used. Line 240 - It was not clear to me how the variation of moisture content led to the observation that soil parameters vary. Line 280 - For FS less than 1 ruptures should occur. Line 348 - Although it is reasonable, the analyzes do not seem to show that the condition of previous moisture content affects the analysis. It seems to me only an opinion. Line 361 – The term "factor of slope safety" should be factor of safety of the slope or slope safety factor. Table 2 – Why not use m/s for hydraulic conductivity? Figure 2 - I suggest that the photos be separated to avoid the impression of continuity between them. Figure 6 - Are the points indicated as small symbols experimental? If there are experimental points should be included along with

the adjustment made.

---

## Referee Comment (RC2) · Anonymous Referee #2 · 21 Aug 2017

The paper addresses a very interesting real case, trying to understand in detail the causes why simple relations between rain intensity and slope stability may be inaccurate for certain regions and conditions. The work is scientifically adequate and the results are interesting regarding the risk management in development countries. Regarding the written document, there is a large number of typing mistakes, signalized in the attached file. In what refers to the references, there is a number of citations not included in the references (see attached file), as well as some references not cited in the text. This topic needs a careful revision before publication. There are also some

mistakes in the cross-references, mainly in the Figures.

Regarding scientific improvements some questions arise: 1 – To determine the SWCC, pressure plates and filter paper was used. Normally, the first equipment is used in drying paths and the second in wetting paths. How the curves were made compatible in the transition points. 2 – It is emphasized in the conclusions the need to perform reliability calculations in future works which the reviewer thinks is an important contribute. In any case, considering the author as intervals for the parameters of each geotechnical horizon, at least some comments could be done regarding the influence of this variability in the FS.

Please also note the supplement to this comment:
https://www.nat-hazards-earth-syst-sci-discuss.net/nhess-2017-242/nhess-2017-242-RC2-supplement.pdf

[revised manuscript text omitted]

---

## Referee Comment (RC3) · Anonymous Referee #3 · 22 Aug 2017

The manuscript describes the application of physically based modelling for the analysis of the effect of anthropic factors to slope stability in in the Campos do Jardao Municipality (Brazil). In order to perform the analysis a detailed geotechnical characterization has been carried out. The manuscript reports an interesting issue that is surely of interest for the readers, anyway some parts of the manuscript are quite surficial and some parts have to be described better to make the general methodology more sound and rigorous. Here below a list of my major comments: a) The introduction should be rewritten in order to focus better on your objectives and the methodology you use. In

my understanding the objective of this work is to assess the stability of slopes considering the effect of the anthropic factors. I would avoid (at least reducing) in the introduction the description of rainfall thresholds since this is not the focus of the paper. I would instead describe state of the art of physically based modelling, moving here the first part of section 2.4. b) The description of state of the art models in section 2.4 is not up-to-date. The references are old. Please have a look to this reference for more recent literature: Rossi, G., Catani, F., Leoni, L., Segoni, S., and Tofani, V.: HIRESSS: a physically based slope stability simulator for HPC applications, Nat. Hazards Earth Syst. Sci., 13, 151-166. c) Please clarify better in the text that SHALSTAB, TRIGERS and so on, are distributed models while the Geostudio Package (SEEP/W and SLOPE/W) makes an analysis at slope scale. d) Which type of method do you use in your stability analysis? e) Concerning the stability analysis you should add a figure with the location of the cuts and loads along your profiles. This is a very important point to be better addressed since it makes your work weak. You know that the loading and unloading of a slope can have different effect on the slope stability depending on the location of the works (J. N. Hutchinson An influence line approach to the stabilization of slopes by cuts and fills Canadian Geotechnical Journal, 1984, Vol. 21, No. 2 : pp. 363-370). Another important issue relates to the shape of landslide surface in the SLOPE/W analysis. Have you drawn your sliding surfaces (the ones in figure 2)? Or have let the software to identify the most critical sliding surfaces? In both cases a figure with the sliding surfaces and their location along the slopes should be added. If possible also a description of the landslides; planar or rotational shape? f) In Table 2 both the effective cohesion and effective friction angles are very high. Please comment on this. g) In table 3 matric suction must have positive values otherwise you should call it pore water pressure

Other minor comments: a) you should explain what is CEMADEN the first time you mention it (Page 2, line 45) b) the sentence at page 5, lines 132-133 is already been written above, please delete it. c) labels in figure 1 are not readable, please modify the figure. d) in caption of Figure 3 you mention deposits of landslide events (blue

cross-hatched areas) but they are not visible in the figure. Please modify the figure.

---

## Editor Comment (EC1) · S. Segoni (Editor) · 27 Aug 2017

Dear authors, all the reviewers I selected have by now posted a comment on the inter-active discussion page.

Although I have to wait the end of the open discussion to formally make my final deci-sion, I think that if you are interested in speeding up the whole revision process, you can feel free to start working at a revised version of the manuscript, according to the three reports of the reviewers.

[Figure]

Regards, S.

---

## Author Comment (AC2) · 17 Oct 2017

Dear editor

In response to the reviewer (Anonymous Referee #2), we have addressed all comments in separate documents. Following reviewers suggestions, we have already began to introduce improvements in the original version of the manuscript.

Looking forwards to hearing from you

[Figure]

Sincerely,

The authors

Please also note the supplement to this comment:
https://www.nat-hazards-earth-syst-sci-discuss.net/nhess-2017-242/nhess-2017-242-AC2-supplement.pdf

**Supplement:**

*We thank the reviewer for the comments and suggestions. Our answers to the reviewer comments are in italic, and the corrections to be included in the new version of the manuscript are in **bold black italic***

a) Regarding the written document, there is a large number of typing mistakes, signalized in the attached file. In what refers to the references, there is a number of citations not included in the references (see attached file), as well as some references not cited in the text. This topic needs a careful revision before publication.

*References were carefully revised as requested. Typing mistakes has been corrected.*

b) There are also some mistakes in the cross-references, mainly in the Figures.

*We have also revised cross-references in the figures.*

Regarding scientific improvements some questions arise:

1 – To determine the SWCC, pressure plates and filter paper was used. Normally, the first equipment is used in drying paths and the second in wetting paths. How the curves were made compatible in the transition points;

*Both methods were used in the drying paths of the samples. In the case of the filter paper method, we followed the recommendations of Marinho and Oliveira (2006). The last paragraph of section 2.3 regarding the derivation of water retention curves was written as:*

> ***Water retention curves -WRC of the residual soils layers were obtained using pressure plate for suctions <100 kPa and filter paper for suctions ≥ 100 kPa.***

*And was modified to*

> ***Water retention curves - WRC of the residual soils layers were obtained using pressure plate for suctions <100 kPa and filter paper for suctions ≥ 100 kPa for the drying path of the samples following the recommendation of Marinho and Oliveira (2006). Results showed that the differences of water retention values at the transition among both method were not significant, making unnecessary further adjustments.***

*Reference: Marinho, FAM; Oliveira, OM (2006) The filter paper method revisited. Geotechnical Testing Journal, ASTM, 29(3):250-258, doi: 10.1520/GTJ14125.*

2 – It is emphasized in the conclusions the need to perform reliability calculations in future works which the reviewer thinks is an important contribute. In any case, considering the author as intervals for the parameters of each geotechnical horizon, at least some comments could be done regarding the influence of this variability in the FS.

*In response to the reviewer comment, line 366 to 368 (Conclusions) of the original version of the manuscript was written as:*

**Future studies should combine modelling tools with probabilistic analysis to consider a wider range of geological-geotechnical and anthropic parameters in the simulations to be able to reproduce more general conditions that occur in the whole municipality.**

*And has been modified to:*

**Since simulations results indicated that the slope safety fact FS was sensitive to the anthropic factors, future studies should combine modelling tools with probabilistic analysis to consider a wider range of geological-geotechnical and anthropic parameters values to be able to reproduce more general conditions that occur in the study area.**

Please also note the supplement to this comment: https://www.nat-hazards-earth-syst-sci-discuss.net/nhess-2017-242/nhess-2017-242- RC2-supplement.pdf

*All recommendations pointed out by the reviewer in the supplemental document will be considered in the updates version of the manuscript.*

---

## Author Response (AR1)

*Our answers to the reviewer comments are in italic, and the corrections to be included in the new version of the manuscript are in **bold black italic***

The combination of experimental laboratory data, field monitoring, geological aspects and the use and occupation of risk areas brings to the study a set of information that deserves to be divulged.

*We thank the reviewer for the comments and suggestions, and we are very pleased to see that our study is of general interest.*

However, some aspects of the study and the text should be evaluated. The following are some comments aimed at improving and clarifying some points of the work:

*We have addressed all comments and clarified several paragraph of the manuscript.*

Line 23 - A reference on the given information of the percentage of the population that lives in urban area in Brazil would be important.

*The reference was included.*

Line 32 – The reference to the removal of forests would not be removal of vegetation? Considering that it is already a deforested area.

*In response to the reviewer comment, line 32 to 33 of the manuscript originally said*

> ***Increased landslide hazard, for instance, has been related to the improper cut-and-fill construction of self-built housing on steep slopes, after the removal of forest cover.***

*And has been modified to:*

> ***Increased landslide hazard, for instance, has been related to the improper cut-and-fill construction of self-built housing on steep slopes, after the removal of vegetation.***

Line 62 – Typo: "Given the lack of detailed data" FROM……..Line 68 - The reference to numerical analysis is very broad. The numerical study was done with the flow analysis, but the stability analysis was done by equilibrium limit, which is not a numerical method.

*In response to the reviewer comment, line 62 of the original version of the manuscript was written as:*

> ***Given the lack of detailed data from historical landslides events in the municipality of Campos do Jordão, the aim of this study was to understand***

*the factors responsible for triggering the landslides of early 2000 in Campos do Jordão using a numerical model of slope stability.*

*And has been modified to (line 62 to 64):*

*Given the lack of detailed data from historical landslides events in the municipality of Campos do Jordão, the aim of this study was to understand the factors responsible for triggering the landslides of early 2000 in the area using a numerical model that fully couple slope stability analysis with saturated/unsaturated transient pore-water pressure simulations.*

Line 107 – Figure 2 needs to be described in the text.

We will included a short description of the landslides typologies in the introduction (line 103 to 107):

*Landslides in the study area are classified as shallow, translational type, with depths of the rupture surfaces less than 2 m. Depending on the position of the rupture, three different kinds of rupture processes are observed: the rupture surface occurs in the residual soil of undisturbed ground; the rupture surface occurs in the residual soil of a slope cut; and the rupture surface occurs in the base of the landfill deposit, or in the slope residual soil with mobilization of the overlying landfill. The last landslide types are more harmful since they mobilized larger amounts of material.*

Line 148 – Typo: Instead of lecture you mean reading?

*Line 148 was rephrased and merged with the next sentence (see below)*

Line 148 to line 150 – This paragraph is repeated.

*The whole paragraph was rephrased and merged with the previous sentence. The original paragraph was:*

*Soil moisture was monitored in the study area at regular intervals of 1 h to a depth 3.0 meters during 2016 using two EnviroScanTM (Campbell Scientific, 2016) probes installed next of the borehole SD-03 (Figure 2). Every EnviroScanTM 133 probes included six capacitance sensors that allowed the determination of soil moisture every 0.5 meter, thus is, at the depths of 0.5, 1.0, until 3.0 m deep. This distribution of depths allowed to monitoring moisture variations for those soil layers which are relevant to this study: landfill, residual soil and saprolite. Sensor calibration was based on the relationship provided by the manufactured (Campbell Scientific, 2016) based on dry and wet readings of each sensor.*

*And now is (line 144 to 149):*

*Soil moisture was monitored during 2016 at hourly intervals and to a depth of 3.0 m using two EnviroScanTM (Campbell Scientific, 2016) probes installed next of the borehole SD-03 (Figure 2). Each probe included six capacitance sensors that measured soil moisture every 0.5 m, thus is, at the depths of 0.5, 1.0, until 3.0 m deep, which allowed to monitor moisture variations of the landfill, residual and saprolite layers. Before the EnviroScanTM capacitance probes were installed in the soil, maximum and minimum values were normalized by matching the raw readings from each sensor at both 0% (held in air) and 100% water levels (submerged in water).*

Line 152 – Instead of Figure 3 it is Figure 4. The figure needs to be described in the text. Line 155 - The phrase seems truncated. The samples were not taken for stability analysis but for the determination of parameters that are used in the analyzes.

*As requested, the sentence was rephrased from:*

*SPT (Standard Penetration Test) boreholes and soil sample collections were performed at six (06) locations along the slope (SD-01 to SD-6) along three different profiles (Figure 3). Disturbed and undisturbed samples were taken under different geotechnical conditions for stability analysis of three (3) critical profiles of the study area (A-A'; B-B'; C-C').*

*To (line 151 to 153):*

*SPT (Standard Penetration Test) boreholes were drilled along three profiles of the study site (A-A'; B-B'; C-C' in Figure 3) at six different positions along the slopes (SD-01 to SD-6, Figure 4). Disturbed and undisturbed samples were taken from the boreholes for the determination of the parameters used for stability analysis.*

Line 154 - The correct spelling is saprolite (saprolith seems to be Greek). This occurs along the text.

*The whole text was corrected*

Line 163 - The sentence is confusing and the order of the essay processes must be rewritten. For example: it is not the application of load that is made with a speed of 0.033 mm/min, but the phase of shearing.

*The original phrase was*

*All samples were saturated and subjected to net normal stress of 25, 50 and 100 kPa applied during 24h with a constant velocity of 0.033 mm/min. The vertical 153 displacements were recorded during this period and after stabilization.*

*And was rephrased to (line 163 to 165):*

> **During the consolidation step, all specimens were saturated for 24h and subjected to net normal stresses of 25, 50 and 100 kPa. Then, in the shearing phase, a constant velocity of 0.033 mm min$^{-1}$ was applied. Vertical and horizontal displacements were recorded during the consolidation and shearing phases.**

Line 181 - The references to figures 3 and 4 are not clear. Where are the boundary conditions?

*It will be included a new Figures (5 and 6) in the text with all boundary conditions used in the seepage and stability analysis.*

Line 184 – It seems to be Figure 5 and not Figure 4.

*The correct reference is Figure 5 rather than Figure 4.*

Line 196 – Why to use a "sandy loam" if the classification used is not from agronomy?

*The soil classification in line 196 was modified from "Sandy loam" to "clayey sand" in accordance with geotechnical classification*

Line 209 – Typo: "…….soils representative of other Brazil…….."??

*The original sentence was:*

> **The values of the resistance and Ksat parameters obtained in this study are comparable to other mean reference values of residual gneiss soils representative of other Brazil**

*And was modified to (line 208 to 210):*

> **In addition, the values of the resistance and Ksat parameters obtained in this study are close to mean reference values of residual gneiss soils representative of other Brazil sites.**

Line 217– Instead of Figure 5 it should be Figure 6.

*The manuscript was corrected, and the sentence makes reference to Figure 9 (lines 217 and 218) rather than figure 5.*

Line 213 - Why not use field capacity?

*Due to the peculiar soil water behavior of tropical oxisols, the definition of field capacity in Brazil do not use the same threshold water potential values of temperate soils (Tomasella et al. 2000, doi:10.2136/sssaj2000.641327x), making comparisons among soils complicated. Therefore, we arbitrary adopted a value of -100 kPa, which*

*approximately corresponds to dry soil conditions, since that threshold highlights the differences among the three soil types.*

Line 219 – The term "humidity" is used as water content. This does not seem correct.

*The sentence was corrected.*

Line 221 – Will the article be published in color? Otherwise references to colors should be retracted from the text and another reference system should be used.

*The publication has no restrictions regarding color figures, so we preferred to keep them as they are for facilitate readers understanding.*

Line 236 - It was not clear to me how the variation of moisture content led to the observation that soil parameters vary.

*Because the probes are installed close enough to assume that they receive the same rainfall amounts, we concluded that differences in the soil moisture behavior should be explained by differences in retention and conductivity properties. The original sentence was:*

> **Contrasting differences in the soil moisture behaviour of the landfill deposit from the probes 3G1 and 3G2 clearly indicates that soil parameters variability is much higher in top layer.**

*And was modified to (line 236 to 238):*

> **Contrasting differences in the soil moisture behaviour of the landfill deposit from the probes 3G1 and 3G2 suggest that the variability of soil parameters is higher in the top layer. This was expected considering that this layer is the result of the cut-and-fill processes mixed with construction wastes of several types.**

Line 274 - For FS less than 1 ruptures should occur.

*As suggested by the reviewer, we modified the sentence in line 274 to:*

> ***"…, where ruptures should occur;…"***

Line 338 - Although it is reasonable, the analyzes do not seem to show that the condition of previous moisture content affects the analysis. It seems to me only an opinion.

*The paragraphs tries to emphasize the fact that, under the influence of leakage, previous rainfall history played a role since the factor of stability is lower previously to the large rainfall event of the end of the simulation period. This can be seen in more clearly in the profile B-B' of Figure 10: in the dry period between day 15 and day 22*

*after the beginning of simulation, it is verified a quick recovery of stability in the simulations that includes the effect of leakage (black curve), which is interrupted with the return of the rainfall. Since this effect was not clearly explained in the manuscript, the text was modified from:*

> **Regarding the rainfall critical values use in early warning system by CEMADEN and the Civil Defense for the Campos do Jordão Municipality, our study showed that, although adequate for the event of 2000, the previous rainfall history played a fundamental role to create conditions favorable to the occurrence of landslides. In other words, the threshold currently used for issue early warning would result in false alarms under initial drier soil conditions.**

*To (line 338 to 344):*

> **Regarding the rainfall critical values use in early warning system by CEMADEN and the Civil Defense for the Campos do Jordão Municipality, although adequate for the event of 2000, our study show that the previous rainfall history, in combination with leakages, played a fundamental role to create favorable conditions for the occurrence of landslides. This is related to the fact that leakages contribute to keep the soil profile closer to saturation at the beginning of the period of more intense rainfall, and consequently the developing of positive pore-pressure conditions. In other words, the threshold currently used for issue early warning would result in late alarms under initial drier soil conditions, at least in heavily disturbed landscapes.**

Line 363 – The term "factor of slope safety" should be factor of safety of the slope or slope safety factor.

*The term "factor of slope safety" was modified to "slope safety factor" in the whole manuscript.*

Table 3 – Why not use m/s for hydraulic conductivity?

*We modified Table 3 and in the current version of the manuscript values are in m/s*

Figure 2 - I suggest that the photos be separated to avoid the impression of continuity between them.

*As suggested by the reviewer, we modified the Figure 2.*

Figure 8 - Are the points indicated as small symbols experimental? If there are experimental points should be included along with Interactive comment on Nat. Hazards Earth Syst. Sci. Discuss.

*Yes, the point in Figure 8 are experimental points. The methods used to obtain them are explained in the last paragraph of section 2.3. Additionally, the caption of figure 8 was expanded to clarify that points indicate measurements.*

**Anonymous Referee #2**

*We thank the reviewer for the comments and suggestions. Our answers to the reviewer comments are in italic, and the corrections to be included in the new version of the manuscript are in **bold black italic***

a) Regarding the written document, there is a large number of typing mistakes, signalized in the attached file. In what refers to the references, there is a number of citations not included in the references (see attached file), as well as some references not cited in the text. This topic needs a careful revision before publication.

*References were carefully revised as requested. Typing mistakes has been corrected.*

b) There are also some mistakes in the cross-references, mainly in the Figures.

*We have also revised cross-references in the figures.*

Regarding scientific improvements some questions arise:

1 – To determine the SWCC, pressure plates and filter paper was used. Normally, the first equipment is used in drying paths and the second in wetting paths. How the curves were made compatible in the transition points;

*Both methods were used in the drying paths of the samples. In the case of the filter paper method, we followed the recommendations of Marinho and Oliveira (2006). The last paragraph of section 2.3 regarding the derivation of water retention curves was written as (line 165):*

> ***Water retention curves -WRC of the residual soils layers were obtained using pressure plate for suctions <100 kPa and filter paper for suctions ≥ 100 kPa.***

*And was modified to (line 166 to 169):*

> ***After saturation soil samples for 12 hours, Water Retention Curves - WRC of the residual soils layers were obtained using pressure plate for suctions <100 kPa and filter paper for suctions ≥ 100 kPa for the drying path of the samples following the recommendation of Marinho and Oliveira (2006). Results showed that the differences of water retention values at the transition among both method were not significant, making unnecessary further adjustments (Figure 8).***

*Reference: Marinho, FAM; Oliveira, OM (2006) The filter paper method revisited. Geotechnical Testing Journal, ASTM, 29(3):250-258, doi: 10.1520/GTJ14125.*

2 – It is emphasized in the conclusions the need to perform reliability calculations in future works which the reviewer thinks is an important contribute. In any case, considering the author as intervals for the parameters of each geotechnical horizon, at least some comments could be done regarding the influence of this variability in the FS.

*In response to the reviewer comment, line 366 to 368 (Conclusions) of the original version of the manuscript was written as:*

> ***Future studies should combine modelling tools with probabilistic analysis to consider a wider range of geological-geotechnical and anthropic parameters in the simulations to be able to reproduce more general conditions that occur in the whole municipality.***

*And has been modified to (line 362 to 365):*

> ***Since simulations results indicated that the slope safety factor FS was sensitive to the anthropic factors, future studies should combine modelling tools with probabilistic analysis to consider a wider range of geological-geotechnical and anthropic parameters values to be able to reproduce more general conditions that occur in the study area.***

Please also note the supplement to this comment: https://www.nat-hazards-earth-syst-sci-discuss.net/nhess-2017-242/nhess-2017-242- RC2-supplement.pdf

*All recommendations pointed out by the reviewer in the supplemental document will be considered in the updates version of the manuscript.*

**Anonymous Referee #3**

*We thank the reviewer for the comments and suggestions. Our answers to the reviewer comments are in italic, and the corrections included in the new version of the manuscript are in **bold black italic***

Here below a list of my major comments:

a) The introduction should be rewritten in order to focus better on your objectives and the methodology you use. In my understanding the objective of this work is to assess the stability of slopes considering the effect of the anthropic factors. I would avoid (at least reducing) in the introduction the description of rainfall thresholds since this is not the focus of the paper. I would instead describe state of the art of physically based modelling, moving here the first part of section 2.4.

*Although we agree with the reviewer that the main focus of the paper is to assess the influence of the anthropic factor on slope stability, we additionally include the implication of those factors in rainfall thresholds to emphasize the impacts on landslide early warning systems, which is the focus of the special issue.*

*We agree with the reviewer that improvements about the state of the art of physical based modeling are necessary. Therefore, the introduction section has been updated to include recent literature.*

b) The description of state of the art models in section 2.4 is not up-to-date. The references are old. Please have a look to this reference for more recent literature: Rossi, G., Catani, F., Leoni, L., Segoni, S., and Tofani, V.: HIRESSS: a physically based slope stability simulator for HPC applications, Nat. Hazards Earth Syst. Sci., 13, 151-166.

*We will include current bibliographical references on stability and flow analysis models. In the introduction section (page 3, lines 65 to 77), the following paragraphs were added:*

> ***Physical-based hydrological models have been widely applied to predict pore-pressure build-up due to the infiltration in shallow landslides (Frattini et al., 2009; Iverson, 2000). Several models, based on the infinite slope concepts, that integrates hillslope hydrology with slope stability, are reported in literature: for instance SINMAP (Pack et al, 1998), SHALSTAB (Dietrich et al., 1998), TRIGRS (Baum et al., 2002) and GEOtop-FS (Rigon et al., 2006).***

> ***During the last decade, physically based landslide prediction models have also been successfully used in early warning systems. Models used in such applications include, among others, the Combined Hydrology and Stability Model-CHASM (Thiebes et al, 2014), the High Resolution Slope Stability***

*Simulator -HIRELESS (Rossi et al, 2013); the SLope-Infiltration Distributed Equilibrium-SLIDE (Liao et al, 2010; Montrasio and Valentino, 2008), the Shallow Landslides Instability Prediction-SLIP (Montrasio 2000; Montrasio et al, 2011). Another slope stability model is the modular software package GeoStudio (2012), in which SEEP/W and SLOPE/W plugins are used to simulate the instability of slopes during extreme rainfalls. Although GeoSlope is a simplified "single slope" model, it has been used in several previous studies to understand the effect of infiltration on rainfall-induced landslides (for instance Ng and Shi, 1998; Gasmo et al., 2000; Kim et al., 2004; Huat et al., 2006; Oh and Vanapalli, 2010; Acharya et al., 2016), producing very good results (Tofani et al., 2006).*

c) Please clarify better in the text that SHALSTAB, TRIGERS and so on, are distributed models while the Geostudio Package (SEEP/W and SLOPE/W) makes an analysis at slope scale.

*We have clarified the difference among the models in the introduction section (page 3)*

d) Which type of method do you use in your stability analysis?

*For stability analysis, we used the Morgenstern & Price method. In order to simulate the transient conditions during the rainfall event of 2000, it was used the module Seep/W. Such cases area analysis by the software GeoSlope an integrated, fully coupled solution. This point has been clarified in the new version of the manuscript in the introduction section.*

e) Concerning the stability analysis you should add a figure with the location of the cuts and loads along your profiles. This is a very important point to be better addressed since it makes your work weak. You know that the loading and unloading of a slope can have different effect on the slope stability depending on the location of the works (J. N. Hutchinson An influence line approach to the stabilization of slopes by cuts and fills Canadian Geotechnical Journal, 1984, Vol. 21, No. 2: pp. 363-370).

A new Figure (5 and 6) has been added in the current version of the manuscript showing the location of cut and fills. It should be emphasized that the results of the present study corroborated the studies of Hutchinson (1984) regarding the effect slope cuts and the location of the loads in the stability factor FS.

Another important issue relates to the shape of landslide surface in the SLOPE/W analysis. Have you drawn your sliding surfaces (the ones in figure 2)? Or have let the software to identify the most critical sliding surfaces? In both cases a figure with the sliding surfaces and their location along the slopes should be added. If possible also a description of the landslides; planar or rotational shape?

*Regarding the sliding surface, simulations were designed to allow Geoslope to identify automatically the most critical rupture surface. To clarify this point, we have added the following paragraph to section 2.4 (line 189 to 191):*

> **All the simulations allowed the slope stability module SLOPE/W to identify the most critical rupture surface (Figure 6). Therefore, the values of the Slope Safety Factor – FS, were the lowest of all conditions analyzed.**

*In addition, a new figure (Figure 6) includes the "slip surface" and the type of processes added showing the location of the type of landslide processes involved (planar rupture).*

f) In Table 2 both the effective cohesion and effective friction angles are very high. Please comment on this.

*To address this comment, the following paragraph has been added in the result section 3.1 (line 206 to 210):*

> **The high values of the resistance parameters shown in Table 2 are associated with the high heterogeneity of the residual gneiss soil, such as the presence quartz particles and other minerals of considerable size in the specimens tested, which confer them high resistance. In addition, the values of the resistance and Ksat parameters obtained in this study are close to mean reference values of residual gneiss soils representative of other Brazilian sites (Costa Filho and Campos, 1991; Ahrendt, 2005; Reis et al., 2011).**

g) In Table 3 matric suction must have positive values otherwise you should call it pore water pressure.

*We adopted the term pore-water pressure in Table 3.*

Other minor comments:

a) You should explain what is CEMADEN the first time you mention it (Page 2, line 39)
*An explanation of the acroname CEMADEN has been added in the introduction section.*

b) The sentence at page 5, lines 144-150 is already been written above, please delete it.

*The whole paragraph has been improved and merged with the previous sentence. The original sentence was:*

> **Soil moisture was monitored in the study area at regular intervals of 1 h to a depth 3.0 meters during 2016 using two EnviroScanTM (Campbell Scientific, 2016) probes installed next of the borehole SD-03 (Figure 2). Every**

*EnviroScanTM 133 probes included six capacitance sensors that allowed the determination of soil moisture every 0.5 meter, thus is, at the depths of 0.5, 1.0, until 3.0 m deep. This distribution of depths allowed to monitoring moisture variations for those soil layers which are relevant to this study: landfill, residual soil and saprolite. Sensor calibration was based on the relationship provided by the manufactured (Campbell Scientific, 2016) based on dry and wet readings of each sensor.*

*And now is (line 144 to 149):*

*Soil moisture was monitored during 2016 at hourly intervals and to a depth of 3 m using two EnviroScanTM (Campbell Scientific, 2016) probes installed next of the borehole SD-03 (Figure 2). Each probe included six capacitance sensors that measured soil moisture every 0.5 m, thus is, at the depths of 0.5, 1.0, until 3.0 m deep, which allowed to monitor moisture variations of the landfill, residual and saprolite layers. Before the EnviroScanTM capacitance probes were installed in the soil, maximum and minimum values were normalized by matching the raw readings from each sensor at both 0% (held in air) and 100% water levels (submerged in water).*

c) Labels in Figure 1 are not readable, please modify the figure.

*Figure 1 has been improved as requested.*

d) In caption of Figure 3 you mention deposits of landslide events (blue cross-hatched areas) but they are not visible in the figure. Please modify the figure.

*The sentence "and deposits of landslides events (blue cross-hatched areas)" was removed in Figure 3.*

---

## Author Response (AR2)

Dear Editor

Many thanks for your helpful comments. Please see below the answers to each topic and the insertions we have made in the manuscript. Our comments are in bold red italic, and the paragraphs included in the text were highlighted in bold black italic.

Comments to the Author:

Dear Authors, I'm sorry to ask you once more "minor revision - review by editor", but I found that, while you addressed properly all referees' comments, you did not considered my comments enclosed to my previous communication (17th of October). Probably you didn't noticed them, so I copy them hereafter, along with a short list of small amendments to the text. I look forward to receive your revised paper, which I believe could make a good contribution to the special issue.

*We did not notice your comments. We apologized for not replying earlier.*

- SMALL CHANGES TO THE TEXT:

L22-23: "In addition, population growth, 22 increased urbanization, recent estimations (CEPED 2012) indicates that..." This sentence is not clear. Please, rephrase.

*The whole paragraph has been improved and rephrase. The original sentence was:*

*In addition, population growth, increased urbanization, recent estimations (CEPED 2012) indicates that more than 160 million inhabitants live in urban areas (about 90% of Brazilian population), and expansion of urban construction into hazardous areas have led to an escalating impact of this natural disaster.*

*And now is (line 22 to 25):*

*Recent studies (CEPED, 2012) indicates that more than 160 million inhabitants live in urban areas (about 90% of Brazilian population), which led to an increased occupation of landslide-prone areas of the urban outskirts.*

L71: HIRESS

*The indicate text was corrected (line 73)*

L2016: are associated = can be explained

*The indicate text was corrected (line 208-209)*

L274-275: "low stability stable". Isn't it better simply "low stability zone"?

*The indicate text was corrected (line 277)*

L421: please link the DOI to the full reference

*The DOI was linked to the reference (line 488 - 490):*

*Segoni, S., Rosi, A., Rossi, G., Catani, F., and Casagli, N.: Analysing the relationship between rainfalls and landslides to define a mosaic of triggering thresholds for regional-scale warning systems. Nat. Hazards Earth Syst. Sci., 14, 2637-2648, 2014. doi:10.5194/nhess-14-2637-2014*

L487: the reference is incomplete

*The reference was completed (line 488 - 490):*

*Segoni, S., Rosi, A., Rossi, G., Catani, F., and Casagli, N.: Analysing the relationship between rainfalls and landslides to define a mosaic of triggering thresholds for regional-scale warning systems. Nat. Hazards Earth Syst. Sci., 14, 2637-2648, 2014. doi:10.5194/nhess-14-2637-2014*

- PREVIOUS EDITOR COMMENTS NOT ADDRESSED:

1- In response to R1L348, you provide an interesting comment on Fig.8 *(now is Fig. 10)*. I think it should be added in the text of the revised manuscript.

*The comment on Fig 10 was included in the section 3.4 (line 303 to 307):*

*Under the influence of leakage, previous rainfall history played a critical role since the factor of stability is lower previously to the large rainfall event of the end of the simulation period. This can be seen in more clearly in the profile B-B' of Figure 10: in the dry period (between day 15 and day 22 after the beginning of simulation) it is verified a quick recovery of stability in the simulations that includes the effect of leakage (black curve), which is interrupted with the return of the rainfall.*

Moreover, you changed "false alarms" in "late alarms": please, check if this change is appropriate.

*In the text the more appropriate terms used were "false alarms" (line 364 to 365). We changed accordingly.*

2- I think the suggestion of R2 at point 2 (influence of variability into FS) should be better addressed in the revised text.

*In response to the editor comment, line 368 to 371 (Conclusions) of the previous version of the manuscript was written as:*

*Since simulations results indicated that the slope safety factor FS was sensitive to the anthropic factors, future studies should combine modelling tools with probabilistic analysis to consider a wider range of geological-geotechnical and anthropic parameters values to be able to reproduce more general conditions that occur in the study area.*

*And has been modified to:*

*Since simulations results indicated that the slope safety factor FS was sensitive to both geotechnical and anthropic factors, future studies of slope stability probabilistic analysis, which takes into account the wider range of parameters values that occur in the study area, are needed.*

3- (something similar was added to the revised manuscript)

*Nothing to add*

4- R3 point e-bis: in my experience it happened a few times that using the lowest value of FS led to an underestimation of the dimensions of the rupture surface. I try to explain with an example: it could happen that the software identifies a small 0.9FS rupture surface and a larger surface with 0.95FS. If you take into account the 0.9 one, you end up with a small slide, while with the 0.95 option you would have a larger slide. In my opinion, it is more correct to take into account the envelope of all theoretical rupture surfaces with FS<1, because the physical meaning of the formula is that everything below 1 will slide. Please, take into consideration this issue.

*In response to the editor comment:*

*The authors agree with these observations. Thus, in this work, the characteristics of the landslides (typology and quantity of mobilized material) obtained from field investigations were considered more important, since they allowed to validate the results of stability analyzes of the profiles studied for the general condition of F.S.<1.*

*In response to your comment, the following sentence was added in line 356 to 362 (Conclusions):*

*Once shallow landslides in the study area usually occur in cut and fill slopes, the rupture surface size and the amount of material mobilized do not vary significantly among events. Therefore, the most useful information for an early warning system perspective is to know whether the value of FS is below 1, regardless how much below that threshold the slope safety factor is. Another relevant information is the timing of the landslide events, since such information is crucial to determine the rainfall thresholds for issuing an early warning. Therefore, information about the rupture surface size, which is essential for assessing potential damages, is beyond the scope of this study.*

---

## Author Response (AR3)

Dear Editor

Many thanks for your helpful comments.

We inform to you that all the corrections have made and insert in the manuscript as highlighted below in red italic bold.

***Please be aware that the publications fees will be covered by another project which was not addressed in the original version of the manuscript. I wander if this can be included in the FunRef tab, since the system is not letting me modified.***

*Acnowledgements*

*Experimental research and field collection were funded by the National Council for Scientific and Technological Development CNPq through the Grant 402240/2012-0. We are also grateful to the National Institute of Science and Technology for Climate Change Phase 2 (CNPq Grant 465501/2014-1, FAPESP Grants 2014/50848-9 and 2015/50122-0, and CAPES Grant 16/2014) for covering publication fees.*

Editor Decision: Publish subject to technical corrections (21 Nov 2017) by Samuele Segoni

Comments to the Author:

Dear Authors, I think your manuscript is finally ready to be published, provided you perform a few very small edits that I recommend you:

L2 (Title): Please check if "Disentalngling" may be more appropriate and grammatically sound than "disentangle".

***The indicate text was corrected (line 2)***

L305: Please change "This can be seen in more clearly" into "This can be seen more clearly"

***The indicate text was corrected (line 303)***

L356: Please change "Once shallow landslides in the study area" into "Since shallow landslides in the study area"

***The indicate text was corrected (line 354)***

Fig. 10: The terms "Instability zone" and "Unstability zone" are used. Please use the same terminology. I suggest to use "Instability".

***The indicate text was corrected in Fig. 10***